# MATRIX FACTORIZATION UNDER THE CONSTRAINT OF CONNECTIVITY BETWEEN OBSERVED AND SOURCE DATA

## MUSCLE SYNERGY ANALYSIS BASED ON CONNECTIVITY BETWEEN MUSCLE AND BRAIN ACTIVITIES

## ABSTRACT

Matrix factorization is a popular method to investigate the hidden elements in observed data for tasks such as speech separation and muscle synergy analysis. The hidden elements may be closely related to the source phenomenon that cause the observed phenomenon. However, conventional methods do not always factorize the observed phenomenon elements with the connectivity between the observed and source phenomena because they only use the observed phenomenon. This paper proposes a matrix decomposition method that constrains the connectivity between observed and source data by using the representations from a decoding model from source data to observed data. The proposed method factorizes matrices by extracting and combining representations and weights from the regression model in the regression process. We applied our method to the corticomuscular system, which is made up of corticospinal pathways between the primary motor cortex and muscles in the body and creates muscle synergies that enable efficient connections between the brain and muscles. In this context, muscle activities are the observed phenomenon and brain activities are the source. Many previous studies have analyzed muscle synergies using only observed muscle activity, but there may be unrevealed muscle synergies under the constraint of the connectivity between brain and muscle activity. We therefore simultaneously recorded the brain activity from multiple regions of an extensive cortical area and the activity of multiple muscles of a monkey's forelimb while it performed a reach and grasp task throughout the course of recovery from a partial spinal cord injury (SCI). Analysis from a dataset of the monkey before SCI showed that some of the muscle synergies calculated from the proposed method using brain and muscle activities, did not exhibit a high degree of similarity to synergies obtained from the conventional method. The proposed method results obtained from the monkey after SCI showed an adaptive change in the number of muscle synergies associated with the degree of functional recovery. Specifically, the numbers of muscle synergies obtained by the proposed method initially increased immediately after SCI and then gradually decreased, while those obtained by a conventional method maintained the same number before and after SCI. These results suggest that our method is able to capture the unrevealed connectivity in the corticomuscular system that contributes to functional recovery: in other words, that it can factorize the observed data under the constraint of the connectivity between the observed and source data. Our work thus demonstrates the importance of using not only observed data but also source data to reveal unknown hidden elements.

## 1 INTRODUCTION

As human beings, we observe a complex mixture of real-world events as phenomena, and each observed phenomenon contains elements of the events in the real-world. For example, the sounds we hear may contain the one voice we are listening to, the voices of other speakers, and various environmental sounds. Separating the elements in the observed phenomena helps us to understand the events individually. Matrix factorization is used to investigate the elements in observed phe-

nomena. For example, a specific speech sound can be extracted from a sound containing a mixture of multiple sounds (Smaragdis, 2007), the appropriate recommendation can be calculated from a customer's purchase history (Li et al., 2006), and muscle synergies can be extracted from multiple muscle activities (Shourijeh et al., 2016).

In some cases, the observed phenomenon is caused by a functional connection with the source phenomenon. This connectivity is a functional connection that causes an observed phenomenon from a source phenomenon, and the factors of connectivity include such as human connections and anatomical connections. In the case of muscle synergy analysis, the corticospinal pathways between the primary motor cortex and muscles function as a corticomuscular system (Liu et al., 2019), and cortical events normally propagate to the peripheral muscles. Studies on humans and monkeys have reported that brain and muscle activity are closely associated with cortico-motoneuronal connections(Lemon, 2008; Baldissera & Cavallari, 1993; Lemon & Griffiths, 2005). In this case, muscle activities are the observed phenomenon and brain activities are the source. The corticomuscular system enables us to perform complex body movements by neurologically combining sets of simpler movements. These sets are observed as the basic pattern of muscle activity, namely, muscle synergies. Muscle activity can be observed by using electromyography (EMG), a technique that reveals bioelectric potential signals. Since the framework of conventional matrix factorization uses only observed data to factorize observed data, it might not factorize the observed phenomenon elements with the connectivity between the observed and source phenomena.

We have therefore developed a method to capture the elements in the observed data by considering the connectivity between both the observed data and the source data. Our basic idea is to use the representations in a deep neural network (DNN) model to predict the observed data from the source data. In this paper, we assume both source and observed data can be measured. Specifically, the factorized matrices can function as an activation scalar value and a vector of weights for the observed data at each sample. We can then obtain the factorized matrix under the connectivity constraints between the observed and source data by extracting these values.

In this paper, we report the results of applying the proposed method to muscle synergy analysis. As stated earlier, the corticomuscular system is composed of corticospinal pathways between the primary motor cortex and muscles (Liu et al., 2019), and we can consider the muscle activities as observed data and the brain activities as source data. To obtain brain activities closely related to muscle activity, we simultaneously measured a monkey's brain signals using both electrocorticography (ECoG) and electromyography (EMG). We measured the muscle and brain activities of a monkey before and after partial spinal cord injury to investigate the potential of the proposed method under the conditions of a stable and a dynamically changing nervous system.

Our main contributions are as follows.

- We show a novel matric factorization framework under the constraint of connectivity between both observed data and source data, in contrast to the conventional approach that uses only observed data.
- We propose a method that utilizes a model representation to predict the observed data from the source data as a factorization matrix.
- We demonstrate the potential of the proposed method to capture unrevealed matrix factors by applying it to muscle synergy analysis with a comparison to the non-negative matrix approach.

## 2 RELATED WORKS

**Matrix Factorization in Statistics:** One approach taken in statistics is to factorize the data by extracting the components that best represent the variation in the data. Principal component analysis (PCA) is typically used for this, where the dimensionality-reduced data representation is a factorized matrix (Lee & Seung, 1999). Another approach is to separate independent components by assuming that the data contains multiple data originating from independent sources. Independent component analysis (ICA) separates signals by independence on the basis of higher-order statistics or temporal correlations (Comon, 1994). The fast fixed-point algorithm for independent component analysis (FastICA) was proposed as a method to improve the convergence of ICA (Hyvärinen & Oja, 2000). Non-negative matrix factorization (NMF) factorizes the matrix and restricts it to be positive

(Lee & Seung, 2000; Wang & Zhang, 2013). Since the real-world events we observe are often composed of linear sums of non-negative values, they have a high affinity with NMF analysis, so it has accordingly been used in a variety of applications (Rajapakse & Wyse, 2003; Smaragdis, 2007; Shourijeh et al., 2016). One study on muscle synergy analysis showed that NMF is the most popular method for analysis (Rabbi et al., 2020). However, these methods factorize the observed data using only the observed data.

**Deep Matrix Factorization:** The deep matrix factorization approach has shown promise in studies that focus on interpretability. Several methods to improve the interpretability by using representations in an autoencoder process as multi-factorized matrices have been proposed (Ye et al., 2018; Song et al., 2015; Wang & Zhang, 2021). To ensure meaning in the multilayered factor matrix, Trigeorgis et al. (2014) proposed a method that pre-trains each layered factor matrix to represent elements such as facial pose, expressions, and identity. Sparseness is also essential to capture the essence of the data. Chang et al. (2021) proposed making the factorized matrix sparse in not only decoding but also encoding. One concept similar to our own is the use of implicit feedback (e.g., purchase history and unobserved ratings) in recommendation systems (Xue et al., 2017). While we feel that such feedback can contribute to improving accuracy, it is not the same as a source phenomenon. And technically, there are few approaches for matrix factorization to use both parameters in the regression process and the weights inside the model.

**Applications:** There are a variety of general applications of the matrix factorization approach. In the biological field, metagenes and molecular pattern discovery have been studied by applying NMF (Brunet et al., 2004; Devarajan, 2008). In the engineering field, image processing tasks such as face recognition (Rajapakse & Wyse, 2003), acoustic processing (e.g., speech-music separation) (Smita et al., 2007), and speech separation (Chien & Chen, 2006) have been studied. As for medical applications, fetal heart rate determination has been explored (Szalai & Mozes, 2014). In the kinematics and kinetics field, muscle synergy analysis is a significant application of matrix factorization Shourijeh et al. (2016). Matrix factorization has also been applied to community structure detection (e.g., social networks, collaboration networks, and citation networks) to study interaction (Ye et al., 2018).

## 3 MATRIX FACTORIZATION UNDER CONSTRAINTS BETWEEN OBSERVED AND SOURCE DATA

Suppose that the observed data are arranged as a matrix $\boldsymbol{Y} = [\boldsymbol{y}_1, \ldots, \boldsymbol{y}_T]^T \in \mathbb{R}^{n \times T}$, which is $n$ channels sampled $T$ times in time series. The observed vector at time $t$ represents $\boldsymbol{y}_t = [y_t^1, y_t^2, \ldots, y_t^n]$. The goal of the proposed method is to approximately factorize the observed data as

$$\boldsymbol{Y} \approx \boldsymbol{V}\boldsymbol{U}. \tag{1}$$

where $\boldsymbol{V} = [\boldsymbol{v}_1, \ldots, \boldsymbol{v}_r]^T \in \mathbb{R}^{n \times r}$ and $\boldsymbol{U} = [\boldsymbol{u}_1, \ldots, \boldsymbol{u}_r] \in \mathbb{R}^{r \times T}$ are the factor matrices to reconstruct the matrix $\boldsymbol{Y}$, and $r$ is the number of factorized components. Suppose the source data is the $m$ channels sampled $T$ times in time series, and the source data at time $t$ is represented as the matrix $\boldsymbol{X}_t = [\boldsymbol{x}_{(1,t)}, \ldots, \boldsymbol{x}_{(m,t)}] \in \mathbb{R}^{m \times T}$, which is $m$ channels sampled $T$ times in time series. Considering the time difference ($\tau$) between observed and source data, the vector of source data at time $t$ consists of data from time $t$ to $t - \tau$. The $i$-th source data vector at time $t$ is $\boldsymbol{x}_{(i,t)} = [x_{(i,t)}, x_{(i,t-1)}, x_{(i,t-2)}, \ldots, x_{(i,t-\tau)}]$.

The basic idea underlying the factorization of matrices under constraints between observed and source data is to factor matrices from a model that includes connectivity constraints between observed and source data. We obtain this model by constructing a DNN prediction model from the source data to the observed data, where the factorized vectors $\boldsymbol{v}$ and $\boldsymbol{u}$ can be assigned as the weight for each channel of observed data and the activation for the weights, respectively.

### 3.1 FRAMEWORK AND ARCHITECTURE

We designed a regression model $\mathcal{M}$ that computes the observed data $\boldsymbol{y}_t$ from the source data $\boldsymbol{X}_t$. As shown in the Fig. 1, the representation in the model is designed to contain activation (scalar) and weights (vector).

Source data $\boldsymbol{X}_t$    Regression model $\mathcal{M}$         Observed data $\boldsymbol{y}_t$

TCNet $(c, r)$ → Mean $(c)$ → Scalarize $(1)$ → Sigmoid $(1)$ → Linear $(n)$

Factors: $u_t$   $\boldsymbol{v}$

Figure 1: Network architecture of proposed method. Activation (scalar) $u_t$ is obtained from the *Sigmoid* layer. Vector $\boldsymbol{v}$ is obtained from the weights in the *Linear* layer.

It is not only the source data that contributes to the factorization at time $t$; data over a certain time range up to $t$ is also considered to be involved in some cases. Temporal convolutional networks (TCN) have outperformed recurrent networks such as LSTM and GRU (Bai et al., 2018) for handling multi-channel time-series data and have been applied to gesture recognition using muscle activity (Rahimian et al., 2021; 2022). Inspired by such research, we utilize TCN to incorporate the time-series features in our method.

**TCNet**: We implement *TCNet* as the TCN in the regression model to capture the features from sequence data. As in (Bai et al., 2018; Rahimian et al., 2021; 2022), the *TCNet* layer consists of a sequence of *TBlock* layers, as shown in Fig. 2. The *TBlock* is the dilated causal convolution, with the dilation factor exponentially increased (1, 2, 4, ...) and the number of filters $c$ and the kernel size $k$. Each *TBlock* consists of two repeated sequences consisting of extended causal convolutions ("DCconv"), weight norm, ReLU and Dropout, and finally a residual connection.

The parameters for *TCNet* are the number of filters $c$, the kernel size $k$, and the number of dilations $d$. When $d$ is set to three, the dilation of TBlock is 1, 2, and 4 sequentially. This layer outputs a matrix $\in \mathbb{R}^{(c \times r)}$ to *Mean* for each input data $\boldsymbol{X}_t$.

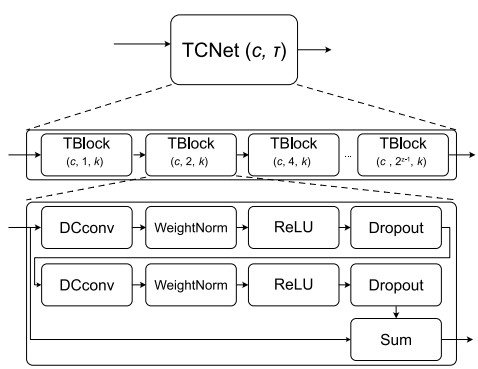

Figure 2: Architecture of temporal convolutional network.

**Mean, Scalarization, and Sigmoid layers**: We designed these layers so that the captured features from *TCNet* would have a positive scalar value (activation $u_t$). The *Mean* layer calculates a vector ($\mathbb{R}^c$) by averaging each filter's result. The *Scalarization* layer is a linear layer to calculate a scalar value from a vector. The *Sigmoid* layer is set to keep the computed scalar value $u_t$ within a positive range. This scalar value $u_t$ is calculated at each time by sliding the window of the source data.

**Linear layer**: The *Linear* layer inputs a positive scalar value and calculates the reconstructed observed vector $\hat{\boldsymbol{y}}_t = [\hat{y}_t^1, \hat{y}_t^2, \ldots, \hat{y}_t^n]$ as

$$\hat{\boldsymbol{y}}_t = u_t \boldsymbol{w}, \tag{2}$$

where $\boldsymbol{w} = [w^1, w^2, \ldots, w^n]$ is the weight for each element of an observed vector. When acquiring a vector of non-negative weights, the weights are clipped so that they are between 0 and 1 during training.

## 3.2 MATRIX FACTORIZATION THROUGH REGRESSION CALCULATIONS

First, the regression model $\mathcal{M}$ is trained using $\boldsymbol{X}_t$ and $\boldsymbol{y}_t$ $t \in (1, \ldots, T)$. Similar to the previous approach, (Shourijeh et al., 2016; Rajapakse & Wyse, 2003), the proposed matrix factorization is verified by the restoration degree of the observed data, so there is no need to separate source and observed data into training and test data. The regression model $\mathcal{M}$ predicts the observed data $\boldsymbol{y}_t$ from the source data $\boldsymbol{X}_t$ as $\mathcal{M}(\boldsymbol{X}_t)$. An activation $u_t$ is obtained from the representation (scalar) from the layer of *Sigmoid*, and a weights vector $\boldsymbol{v}$ is obtained by extracting the weights at the *Linear*

layer in a model $\mathcal{M}$. $u_t$ is the activation of the component at time $t$. $\boldsymbol{X}_t$ is a sample extracted from the time series source data with a $\tau$-width time window. When the window locates at $t = 1$, $\boldsymbol{X}_1$ is extracted from the time series source data. Regression model $\mathcal{M}$ inputs $\boldsymbol{X}_1$ and outputs $\boldsymbol{y}_1$, and $u_1$ is obtained from this calculation process. By sliding the window, regression model $\mathcal{M}$ inputs $\boldsymbol{X}_t$, $t \in (1, \ldots, T)$ and outputs $\boldsymbol{y}_t$, $t \in (1, \ldots, T)$. $u_t, t \in (1, \ldots, T)$ is obtained from each time. By concatenating $u_t$ from each time, we get the activation vector $\boldsymbol{u} = [u_1, u_2, \ldots, u_T]$. In this way, we can obtain an activation and a weight vector of a component.

Algo. 1 explains the factorization procedure. The reconstructed observed data $\hat{\boldsymbol{y}}_t$ can be more accurately reconstructed by ensembling $\mathcal{M}$ as

$$\hat{\boldsymbol{y}}_t = \sum_{i=1}^{r} \mathcal{M}_i(\boldsymbol{X}_t). \qquad (3)$$

where $r$ is the number of factorized components and $\mathcal{M}_i$ is the $i$-th model. In computing matrix factorization, the number of components $r$ needs to be given. The appropriate number of factorized components $r$ is determined according to evaluation indices such as the degree of restoration. $\mathcal{M}_i, i \in (1, \ldots, r)$ are trained in parallel to minimize the objective function using the same dataset across models. From each $\mathcal{M}_i$, we obtain the activation vector $\boldsymbol{u}_i$

---

**Algorithm 1** Factorization procedure

Input: $\boldsymbol{X}_t$ and $\boldsymbol{y}_t, t \in \{1, \ldots, T\}$, and $r$.
**repeat**
    Update parameters in $\mathcal{M}_i, i \in \{1, \ldots, r\}$
    Clip $\boldsymbol{w}$ in *Linear*: $0 < \boldsymbol{w} < 1$
**until** objective function is converged
$\boldsymbol{u}_i \Leftarrow [u_t], t \in \{1, \ldots, T\}$ from *Sigmoid* in $\mathcal{M}_i$
$\boldsymbol{v}_i \Leftarrow \boldsymbol{w}$ from *Linear* in $\mathcal{M}_i$
$\boldsymbol{V} \Leftarrow [\boldsymbol{v}_i], i \in \{1, \ldots, r\}$
$\boldsymbol{U} \Leftarrow [\boldsymbol{u}_i], i \in \{1, \ldots, r\}$
Output: $\hat{\boldsymbol{Y}} = \boldsymbol{V}\boldsymbol{U}$

---

and the weight vector $\boldsymbol{v}_i$. The factorized matrix $\boldsymbol{V}$ is obtained by combining the weight vectors as $[\boldsymbol{v}_1, \ldots, \boldsymbol{v}_r]$. The factorized matrix $\boldsymbol{U}$ is also obtained by combining the activation vectors as $[\boldsymbol{u}_1, \ldots, \boldsymbol{u}_r]$

The objective function is

$$\min \mathcal{L}(\boldsymbol{y}_t, \sum_{i=1}^{r} \mathcal{M}_i(\boldsymbol{X}_t)) + \mathcal{R}(\boldsymbol{w}) \qquad (4)$$

where $\mathcal{L}(\boldsymbol{y}_t, \sum_{i=1}^{r} \mathcal{M}_i(\boldsymbol{X}_t))$ is the reconstruction loss and $\mathcal{R}(\boldsymbol{w})$ is the regularization term for the weight. The parameters of the models are updated to minimize the objective function are denoted as $\mathcal{M}_i$. At each iteration of learning model, each element of the weights is clipped from 0 to 1 to obtain the positive factors of the matrix. The weight vector $\boldsymbol{w}$ in $\mathcal{M}_i$ is used as $\boldsymbol{v}_i$. $\boldsymbol{V}$ and $\boldsymbol{U}$ are obtained by concatenating of vectors $\boldsymbol{v}_i$ and $\boldsymbol{u}_i, i \in \{1, \ldots, r\}$, respectively. Reconstructed observed data $\hat{\boldsymbol{Y}} = [\boldsymbol{y}_t], t \in \{1, \ldots, T\}$ is reconstructed by $\boldsymbol{V}$ and $\boldsymbol{U}$.

## 4 EXPERIMENT

We applied the proposed method to muscle synergy analysis. As stated in the introduction, our bodies contain corticospinal pathways between the primary motor cortex and muscles that function as a corticomuscular system (Liu et al., 2019). Normally, cortical events propagate to the peripheral muscles, and the corticomuscular system enables us to perform complex body movements by neurologically combining sets of simpler movements. These simpler movements are observed as the basic pattern of muscle activity, namely, muscle synergies. Muscle synergy analysis is designed to capture this efficient neural connection between the brain and muscle. However, the conventional approach uses only muscle activities (observed phenomena) to capture the muscle synergies, and there may still be unexplored muscle synergies that remain hidden. In applying the proposed method to muscle synergy analysis in this experiment, we consider the muscle activities as the observed phenomenon and the brain activities as the source phenomenon. To test the proposed method, we measured the muscle and brain activities simultaneously in a monkey.

### 4.1 DATA MEASUREMENT

We measured the biopotential signals derived from the muscles and brain activities. The invasively measured electrocorticographic (ECoG) signal was selected as the brain activity data because it

contains more detailed brain activity than a non-invasively measured signal. Thus, to obtain both the brain activity and the muscle activity, we recorded ECoG and electromyographic (EMG) signals simultaneously from the contralateral hemisphere and the ipsilateral forelimb muscles of a monkey, respectively.

The monkey used in this experiments was an adult Japanese macaque monkey (Macaca fuscata; Monkey D: female, 6.0 kg, 7 years old). The Ethical description is in A.1.

**Processing**: ECoG signals were down-sampled to 500 samples per second, similar to previous works (Shin et al., 2012; Hidenori et al., 2012). Fourth-order Butterworth bandpass filters were applied to each ECoG signal, dividing them into multiple specific bands ($\delta(1.5 - 4Hz), \theta(4 - 8Hz), \alpha(8 - 14Hz), \beta1(14 - 20Hz), \beta2(20 - 30Hz), \gamma1(30 - 50Hz), \gamma2(50 - 90Hz), \gamma3(90 - 120Hz)$ and $\gamma4(120 - 150Hz)$). The nine bandpass filters split each of the 15-channel ECoG signals into nine signals to produce 135 channels of bandpass-filtered signals. We performed normalization and smoothing the same as in (Shin et al., 2012; Hidenori et al., 2012). A bandpass filter that passes a 20–500-Hz frequency was applied to the EMG signals for removing motion artifacts. Bandpassed signals were then rectified and passed through a fourth-order low-pass filter with a cut-off frequency of 4 Hz. Finally, the signals were down-sampled to 500 Hz. To account for the nerve signal transmission delay between brain and muscles, delayed signal values is also used. The value of discrete-time step-size $\Delta t$ is set to 20 ms as in the previous study (Shin et al., 2012). The muscle activity at time $t$ is predicted using 10 time points starting 200 ms before the target time t. The $i$-th ECoG data vector at time $t$ is $\boldsymbol{x}_{(i,t)} = [x_{(i,t)}, x_{(i,t-20)}, x_{(i,t-40)}, \dots, x_{(i,t-200)}]$, and the ECoG data matrix at time $t$ is $\boldsymbol{X}_t = [\boldsymbol{x}_{(1,t)}, \dots, \boldsymbol{x}_{(135,t)}]$.

**Baseline**: We utilized non-negative matrix factorization (NMF) as a baseline and used the Python library scikit-learn. All parameters were set to the same as those for the pre-set.

**Metric**: Variance accounted for (VAF) is the main metric of the degree of reconstruction and used to determine the number of synergies (Steele et al., 2013; Delis et al., 2013), where the dimension of factors is determined when VAF exceeds 0.9. The dimension of factors is used as the synergy number in some cases (Hug et al., 2010; Frère & Hug, 2012; Chvatal & Ting, 2013). The specific calculation is decribed in A.4. In our analysis, the similarity of the muscle synergy components resulting from the two methods (proposed and NMF) is calculated by the cosine similarity of the weights of the synergies.

**Model selection**: We performed three times optimization and adopted the model with the largest VAF because the VAF of the proposed method varies slightly each time.

## 4.2 MUSCLE SYNERGY ACQUISITION POTENTIAL USING ONLY MUSCLE ACTIVITY (CONVENTIONAL FRAMEWORK)

First, we examined the proposed method's availability of muscle synergies by using only muscle activities (conventional framework). The muscle synergies were determined based on the degree of reconstruction (VAF) and the number of synergies. Theoretically, as the number of synergies increases, the VAF becomes higher because the reconstruction accuracy improves. Researchers have shown that the VAF of NMF increases with the increase of the number of muscle synergies in the case of using only muscle activities (Rabbi et al., 2020; Gui & Zhang, 2016). If the proposed method can calculate muscle synergies by factorizing the input matrix, the VAF should increase with the number of muscle synergies. We tested the VAF of the proposed method to see if it would increase and exceed 0.9, as the number of synergies increases only when using muscle activity. The proposed method inputs an EMG vec-

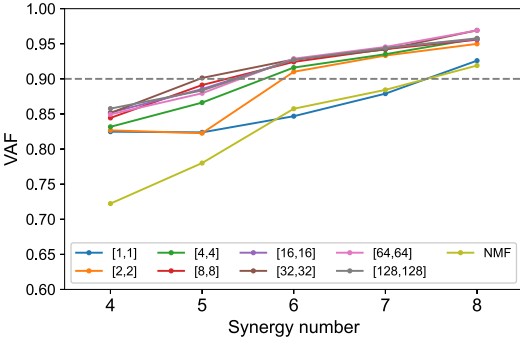

Figure 3: VAF of proposed and NMF methods for each number of synergies. In the proposed method, we experimentally explored the number of filters from [1, 1] to [128, 128].

tor $\boldsymbol{y}_t = [y_t^1, \ldots, y_t^{14}]$, and the output is also supposed to be $\boldsymbol{y}_t$ without changing the architecture as following

$$\hat{\boldsymbol{y}}_t = \sum_{i=1}^{r} \mathcal{M}_i(\boldsymbol{y}_t). \tag{5}$$

We used data from a one-day experiment that included 90 trials in the intact state (no neurological disorders). In the proposed method, the number of filters $c$ of the temporal convolutional networks in the activation layer is a key parameter to determine the potential of synergy acquisition. Thus, we experimentally tested eight types of filters here: [1, 1], [2, 2], [4, 4], [8, 8], [16, 16], [32, 32], [64, 64], and [128, 128]. The vector length of the number of filters is the number of dilations in the temporal convolutional network, which is two in all eight types. For example, in the case of [8, 8], the first "8" is the number of filters at the first dilation, and the second "8" is the number of filters for the second dilation.

Figure 3 shows the VAF results of the proposed method (eight different numbers of filters) and NMF by increasing the synergy number for the same data. Most of the VAF results calculated by the proposed method increased with the synergy number, as did those of NMF. The dashed horizontal line indicates a VAF of 0.9, which determines the muscle synergy number. For the same number of muscle synergies, VAF tended to be higher for a larger number of filters. By confirming that VAF increases with increasing the number of synergies, we verified the possibility that the proposed method can analyze muscle synergies.

### 4.3 NUMBER OF FILTERS

We also tested whether the proposed method can acquire muscle synergy under connectivity constraints between brain and muscle activity. Fig. 4 shows the VAF per muscle synergy number for the proposed method and NMF. Most of the proposed method's VAF values increased with the number of muscle synergies not depending on the number of filters. The higher number of filters tended to make VAF higher.

We can see that the determined number of muscle synergies, where VAF exceeds 0.9 at first, depends on the number of filters. At the numbers of filters [32, 32] and [64, 64], the determined number of muscle synergies is eight. At the number of filters [128, 128], the determined number of muscle synergies is six. Defining the number of filters depends on how the muscle synergies are used. To compare the components of muscle synergies from the proposed method and NMF, we set the number of filters to [32, 32], which is the minimum number of filters when the number of muscle synergies is eight.

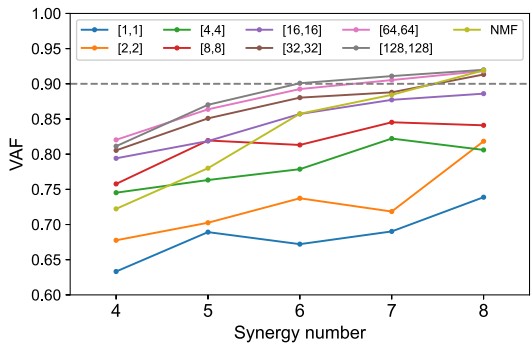

Figure 4: Muscle synergy analysis by factorization using brain and muscle activities in the proposed method.

### 4.4 COMPARISON OF MUSCLE SYNERGY COMPONENTS

We examined whether the proposed method can capture the unrevealed muscle synergy components that are difficult for the conventional approach to extract. As a muscle synergy component, a vector of weights and activations is obtained from a muscle synergy. A, B, and C in Fig. 5 respectively indicate the eight components from the proposed method using only EMG, NMF, and the proposed method using ECoG and EMG. The left panels in the figure show the weight of the muscles of each synergy, while the right panels show the activation of each synergy, which is calculated by averaging all trials. To compare the similarity of each component, we calculate the similarity matrix. D in Fig. 5 shows the similarity matrix using the weights from these three approaches. The upper panel in D shows the similarity matrix of the weights between the proposed method (only using EMG) and the NMF method, while the lower panel in D shows that between the proposed method

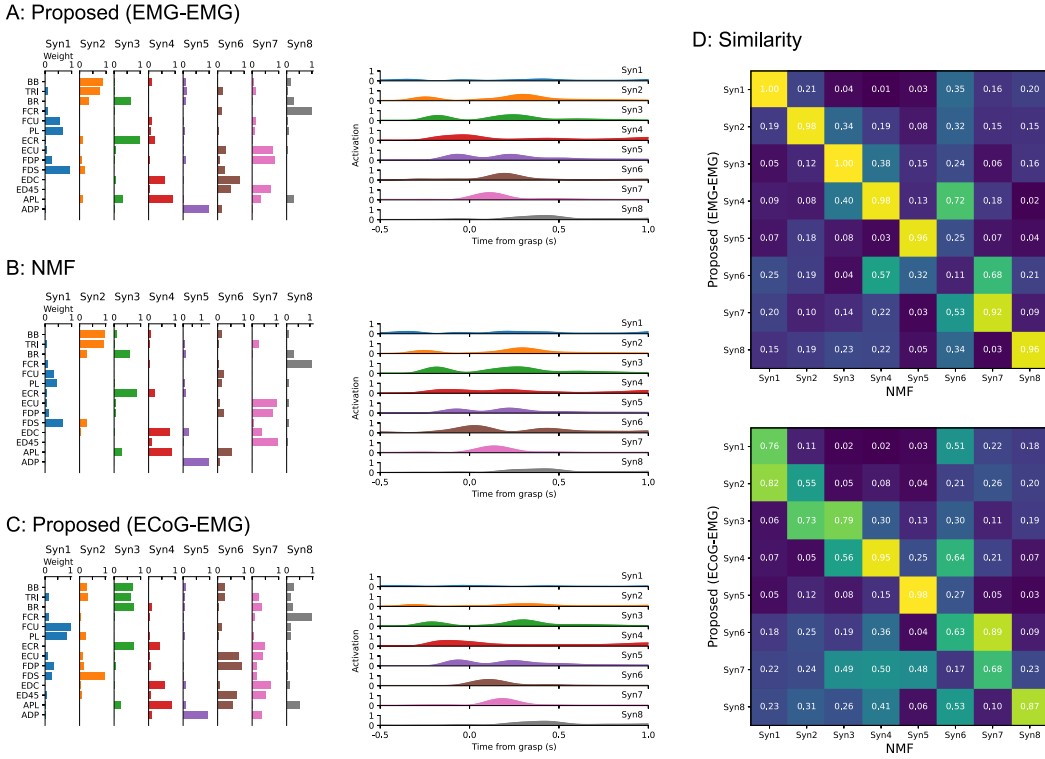

Figure 5: Weights and activations of muscle synergy components from the proposed method (using only EMG), NMF (using only EMG), and the proposed method (using EMG and ECoG).

using ECoG and the EMG and NMF methods. Using only EMG data, the similarity of almost all components of muscle synergy was close to 1, meaning that they were all identical for the proposed and NMF methods. The averaged computation times of the proposed method (basically using GPU: NVIDIA RTX3090) and NMF (CPU: AMD Ryzen Threadripper 3970X) are 6209.3 sec and 12.0 sec, respectively. The computational complexity of the proposed method is higher than that of the NMF. The low similarity of synergies of 6 in both the proposed method (using only EMG) and NMF may have been due to the high degree of freedom of activation in the proposed method. As a result, synergies 4 and 7 of the proposed method represent the corresponding synergies in NMF, while also representing synergy 6 in NMF by the cooperative activation of the two synergies. It may be possible to solve this problem by setting a constraint term such that the activation acts independently between the synergies.

Some of the similarities between the proposed components (using ECoG and EMG) and NMF did not exhibit a high degree of similarity. For example, the similarities of the proposed method's synergy 1 and 2 to synergy 1 in NMF were 0.76 and 0.82, respectively. The results indicate that the proposed method separates synergy 1 of NMF into two synergies. These separated components may be the result of capturing the connectivity between brain and muscle activities. The same can be said for synergy 2 and 3 of the proposed method.

## 4.5 ADAPTIVE CHANGE ANALYSIS

We next investigated whether the proposed method can capture the adaptive change of muscle synergy components during nervous recovery. First, brain and muscle activity was measured from an intact (no neurological disorders) monkey. Then, we performed partial SCI surgery on the monkey and measured its activity for 30 days (not every day) after the surgery. We explored the number of components of each of the experiment days in the proposed and NMF methods.

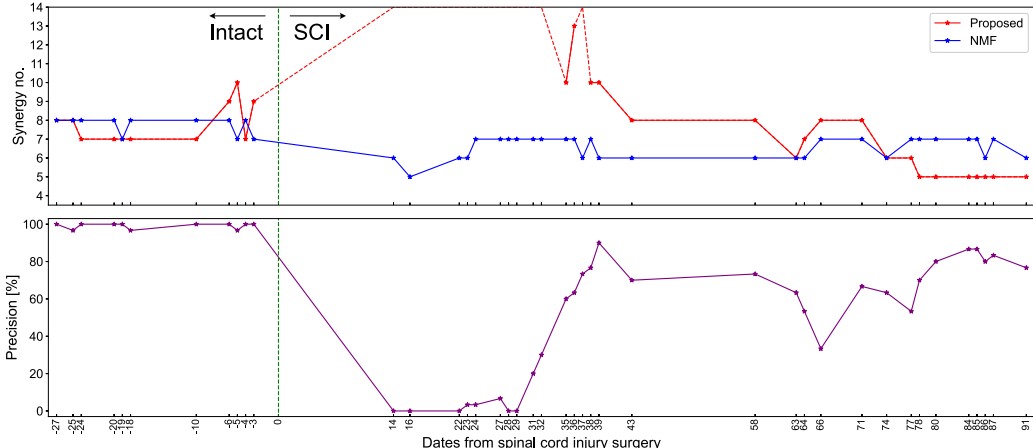

Figure 6: Number of components (upper panel) and task precision (lower panel) in each day. The numbers of components of the proposed and NMF methods are plotted in red and blue lines, respectively, in the upper panel. The days depicted by the red dashed lines are those for which no muscle synergy was computed below 14. The vertical dashed green line indicates the SCI surgery day.

Figure 6 shows the number of components (upper panel) and task precision (lower panel) in each day. We can see the precisions during the intact period were almost 100 %, then moved to nearly 0 % for a month after the surgery, and then improved to around 80 % after three months. These results suggest that the monkey performed the task before the surgery based on the pathways between the brain and muscles acquired so far. Immediately after the partial SCI surgery, it was difficult for the monkey to use the pathways because many of the previously acquired pathways were disconnected and it was more difficult to move the upper limb. The monkey regained the function to perform the task by adaptively changing the pathways using the residual nervous connections, and improved the precision of the task. The numbers of components of the proposed and NMF methods are plotted in red and blue lines, respectively. The days dashed in red (proposed) are days where muscle synergy was not calculated below 14 (the number of measured muscles).

The numbers of components of NMF's muscle synergy were stable both before and after SCI (Fig. 6). While those of the proposed method were similar during the intact period, they exceeded 14 about a month after the surgery and then gradually decreased and converged around three months after the surgery. We presume that the number of components increased immediately after the surgery because the monkey could not use the pathways obtained so far, and had to use the residual pathways (not sophisticated). The number of components decreases and converges in the adaptive change of the nervous system, so the transition of the proposed method's results might be explainable from the neurological point of view. This result indicates that the proposed method can potentially be used to analyze muscle synergy with adaptive nervous change, which is difficult to extract in the conventional approach.

## 5  CONCLUSION

In this paper, we proposed a matrix factorization method under the constraint of connectivity between observed and source data. The core idea of our method is to use the representation in a model that predicts the observed data from the source data as a factored matrix. Applying the proposed method to muscle synergy analysis using a monkey's brain and muscle activities demonstrated its potential to reveal unknown muscle synergy components, which are difficult for the conventional approach to capture. Specifically, the proposed method's transitions of the number of components after partial SCI surgery indicated the adaptive changes in the corticomuscular system. Through two analysis experiments, we demonstrated the importance of using not only observed data but also source data to extract the elements hidden in observed data, and showed that the proposed method's potential to capture these hidden elements.

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

## A   APPENDIX

### A.1   ETHICS

All animal experimental procedures were performed in accordance with the guidelines of the National Institute of Health and the Ministry of Education, Culture, Sports, Science, and Technology of Japan, and were approved by the Institutional Animal Care and Use Committee of the Tokyo Metropolitan Institute of Medical Science (Approval No. 18034). The monkey was monitored closely, and animal welfare was assessed daily or, if necessary, several times a day.

### A.2   ECoG AND EMG SIGNALS

We implanted a 15-channel grid electrode array in which the diameter of each electrode was 1 mm and the inter-electrode distance was 5 mm (Unique Medical, Japan) beneath the dura mater. To implant the grid electrode array on the cortical surface over the sensorimotor cortex, the cortices around the arcuate sulcus and the central sulcus on the right side were exposed by a craniotomy. As to the muscle activity, electrodes were implanted in 14 muscles of the left forelimb: three elbow muscles [triceps brachii (TRI), biceps brachii (BB), brachialis (BR)], five wrist muscles [extensor carpi radialis (ECR), extensor carpi ulnaris (ECU), flexor carpi radialis (FCR), flexor carpi ulnaris (FCU),

palmaris longus (PL)], five digit muscles [extensor digitorum communis (EDC), extensor digitorum 4,5 (ED45), flexor digitorum superficialis (FDS), flexor digitorum profundus (FDP), abductor pollicis longus (APL)], and one intrinsic hand muscle [adductor pollicis (ADP)]. ECoG and EMG signals were recorded simultaneously using a CerebusTM data acquisition system (BLACKROCK MICROSYSTEMS, Utah, USA) at a sampling rate of 2,000 Hz.

### A.3 BEHAVIOR TASK

The monkey performed the task of reaching and grasping for a piece of food multiple times in one experimental day. On each day, the decision to perform the measurement experiment and the number of trials were determined on the basis of the monkey's condition. In total, we conducted the measurement experiment for 41 days. Each recording session consisted of ˜100 trials.

### A.4 VAF CALCULATION

Variance accounted for (VAF) is a metric of the degree of reconstruction. This calculation uses true value vectors $\boldsymbol{y}^m = [y_1^m, \ldots, y_T^m], m \in (1, \ldots, 14)$ and reconstructed vectors $\hat{\boldsymbol{y}} = [\hat{y}_1^m, \ldots, \hat{y}_T^m], m \in (1, \ldots, 14)$, where $m$ is the number of measure muscles. The VAF is calculated as following

$$VAF = 1 - \frac{\sum_{m=1}^{14} Var(\boldsymbol{y}^m - \hat{\boldsymbol{y}}^m)}{\sum_{m=1}^{14} Var(\boldsymbol{y}^m)}, \quad (6)$$

where $Var(\boldsymbol{y}^m - \hat{\boldsymbol{y}}^m)$ is the variance of $\boldsymbol{y}^m - \hat{\boldsymbol{y}}^m$, the square of the standard deviation. $Var(\boldsymbol{y}^m)$ is the variance of $\boldsymbol{y}^m$.

