# OpenReview forum: "Matrix factorization under the constraint of connectivity between observed and source data ~ Muscle synergy analysis based on connectivity between muscle and brain activities ~"
_ICLR.cc/2023/Conference — Submitted to ICLR 2023_

### Official Review · Reviewer_VHFL · 2022-10-24

**Confidence:** 3
**Correctness:** 2
**Technical Novelty And Significance:** 2
**Empirical Novelty And Significance:** 1
**Recommendation:** 3

**Clarity, Quality, Novelty And Reproducibility:**

The manuscript writing was adequate, although some of the modeling details could have been presented more clearly.  Its not clear to me how the number of parallel TCN models (parameter ‘r’) were trained.

I believe the model as applied here is novel, at least in its application.

As noted above, I found more details on the model training to be needed to reproduce the results.


**Strength And Weaknesses:**

Strengths:
(1) Muscle synergy analysis is widely used, but is based on a fairly ad hoc statistical approach and so new approaches could be very impactful.
(2) The model appears technically sound, and the comparisons provided with NMF and across filter numbers are worthwhile.

Weaknesses
(1)  The framing of synergies and their neuroscientific context is somewhat lacking. The premise of the paper is that muscle synergies can be predicted from the cortical inputs, e.g. We applied our method to the corticomuscular system, which is made up of corticospinal pathways between the primary motor cortex and muscles in the body and creates muscle synergies that enable efficient connections between the brain and muscles.”, however synergies are thought to be generated in the spinal cord and some of their first characterization was in frogs, a species without a motor cortex.
One reason this is problematic is that the paper is framed as revealing *new* synergies, e.g. “However, the conventional approach uses only muscle activities (observed phenomena) to capture the muscle synergies, and there may still be unexplored muscle synergies that remain hidden” However, based on the model design it seems like it should be detecting a subset of the muscle-only synergies. Moreover, synergies is largely defined in a muscle-centric way. It is certainly the case that discovering cortico-muscular shared synergies is interesting, but the framing is very different.
(2) There are no details provided on the TCN training, and importantly how data was split up for train and test splits. This is especially important for the SCI experiments.
(3) There are multiple alternative models that could be considered, for instance performing NNMF on a linear or nonlinear prediction of EMG from ecog activity. The specific motivation for the increased number of parameters and model structure of the TCN is not provided. One of the appeals of NNMF is its simplicity, allowing it to be used across paradigms and contexts, and this TCN introduces a lot of added complexity, with only minimal gains in VAF at high numbers of syneries.
(4) For the SCI experiments – there is no ground truth present and so it is impossible to evaluate which technique is ‘correct’. As noted above, without knowledge of how much data is required for model training, it is hard to know if the increase in number of synergies observed is a result of

Nits:
-	‘connectivity’ is misleading, as it isn’t using the structural connections between the brain and body.
-	Figure 6: would help to have an estimate of variance for the number of synergies, e.g. from using different subsets of data to train/test.


**Summary Of The Paper:**

This paper introduces a deep non-negative matrix factorization approach for time series that offers the ability to constrain the factorization to common factors between a source and a receiver. They apply this approach to the detection of muscle synergies detected by EMG at the forearm of a monkey performing a reach and grasp task, with and without the constraint of being driven by ecog activity from the sensorimotor cortex. The new model is compared to traditional non-negative matrix factorization, a gold standard for synergy detection in the field. The proposed model works by processing an input timeseries with a variable number of parallel stacked TCN networks and scalarizing and normalizing each networks output. Compared to traditional NMF analysis the proposed approach demonstrates improved variance accounted for (VAF) for low numbers of synergies, and recovers comparable number of synergies from EMG alone. When using the ecog as the input to the TCN model, there are more differences in the recovered synergies. Finally the later approach using the TCN input is compared to NMF for synergy detection after inducing spinal cord injury, which shows large differences in detection.

**Summary Of The Review:**

Overall I found the TCN model presented somewhat interesting and potentially novel and I believe the problem of synergy detection itself is important. However I found the framing and goal of the paper somewhat misguided from a neuroscience bent, and a lack of alternative models discussed. It is possible that a revision could alleviate some of these concerns, but this would require a fairly thorough rewrite of problematic neuroscientific framing and a much deeper consideration of alternative architectures or modeling approaches to prove the importance of the one chosen.

---

### Official Review · Reviewer_jF4J · 2022-10-25

**Confidence:** 5
**Correctness:** 3
**Technical Novelty And Significance:** 2
**Empirical Novelty And Significance:** 2
**Recommendation:** 5

**Clarity, Quality, Novelty And Reproducibility:**

The paper is relatively clear in terms of the background information and goal description. However, the technical contribution is not clear and the space devoted to describe the technical challenges is very limited. There are many areas that need polishing, especially with respect to the modeling challenges that the paper solves and the evaluation choice for the predictions.

**Strength And Weaknesses:**

With respect to its strengths, the paper is well organized, the problem is relevant to the ML community and the neuroscience community. The paper provides a very didactic description of the proble, although the description of the contribution is rather limited. Another strength of the paper is that the evaluations consider various perspectives of the problem. However, the measure used for evaluating performance provides a limited perspective about the performance of the models.
That being said, there are some items that need some work. The description of the factorization is not very rigorous. The factorization of $Y$ is rigorous but the factorization that incorporates $X$ is not formally introduced. The factorization itself is an application of ideas of Bai et al. (2018) Rahimian et al. (2021; 2022) to the reconstruction of observed data $Y$ and other ideas. The main novel component of this work is the application to modeling corticomuscular synergies. The evaluation is interesting but there is still work that could be done. In particular VAF is defined but it is not clear what the ground truth or the reference for the variance comes from.

**Summary Of The Paper:**

This paper presents a matrix factorization technique that incorporates also information from a, so called, source data in addition to the data from the direct observations. The paper’s authors motivate the challenge by focusing the discussion on the application of the method to corticomuscular system with brain activity associated to several muscles of a monkey’s forelimb (reach and grasp tasks) during recovery of spinal cord injury. The model in itself if constructed with a DNN prediction model that links the $T$-dimensional source data with $m$ channels with the observed $T$-dimensional observational data with $n$ channels. The architecture of the DNN include: a TCNet, mean layer, scalarization layer, a sigmoid layer, and finally a linear layer. The source data is fed to the TCNet, the factorization DNN reconstructs the observed data where the sigmoid and linear layers provide the factors of the procedure.  The evaluation on the data from the corticomuscular system aimed to reconstruct the “synergies” between the brain and the muscle. This evaluation first considers only muscle activity data, then uses brain and muscle activity, and analyses the synergies identified and the changes identified before and after the SCI surgery.

**Summary Of The Review:**

The paper has its pros, such as the relevance and the potential applicability. However, there are several reservations I have with respect to both the clarity of the technical contribution, the modeling, and the the choice of performance measure. More work is needed to polish the description of the contribution and the evaluations in order to have a realistic assessment of the quality of this work. I detailed my comments on what could be improved in the Strength and Weaknesses section.

---

### Official Review · Reviewer_89Ck · 2022-10-30

**Confidence:** 4
**Correctness:** 3
**Technical Novelty And Significance:** 2
**Empirical Novelty And Significance:** 2
**Recommendation:** 3

**Clarity, Quality, Novelty And Reproducibility:**

The writing clarity could be improved, as described above not all the details of the proposed approach are immediately clear. In general, presentation of the proposed regression model with non-negativity constraint on the weights of the output linear layer as a novel matrix factorization method is highly misleading. It would have been clearer is the description was closer to the above. In general, the paper contains a wide gap between the claimed and the actual novelty. It does not look like the data is publicly available to warrant reproducibility of the applications. I am not sure if the authors plan to release the code and data.


**Strength And Weaknesses:**

# Strengths
- The paper is mostly clearly written apart some undefined notions and at times lack of explanations.
- The application is interesting and potentially extensible to other fields and situations. In a way, the paper proposes an autoencoder with deep convolutional encoder and linear decoder, only this architecture is not auto-restoring the input, but predicts the output via reconstruction loss, i.e. what in deep learning is called a regression task.
# Weaknesses
- My main concern is the lack of technical novelty. It is a solid application paper, where a regression model is used to interpret latent factors of one of the signal modalities.
- The writing of the paper is sometimes confusing.
  - For example, the introduction talks about "connectivity", but it is unclear what kind of connectivity is implied, anatomical, functional, hypothetical in some implicit data manifold, or something else. The meaning of the "connectivity" does not become clear further.
  - The Introduction could clarify that both the source and the measured data are actually measured. This is to avoid the confusion with matrix factorization models, where sources commonly denote latent factors (e.g. in ICA).
  - Section 3 around expression 1 could be clearer if the fact that the model operates on time-windows would have been made explicit.
  - Algorithm 1 needs to specify $\mathbf{Y}_t$ as an input as well to be trained as a regression model.
- It would be informative to compare computational complexity of the proposed approach and NMF. The results correlate highly (Figure 5).


**Summary Of The Paper:**

Alternatively to the common way of finding latent factors in the data via a matrix factorization, the paper proposes to interpret a deep regression model with a linear layer as a matrix factorization of the output data. The main intention of this re-interpretation is to be able to interpret the soft-maxed input to the final linear layer as the weights, and the linear layer weights as the features.


**Summary Of The Review:**

An interesting application paper with a misleading narrative that inflates the novelty of the proposed modification of a deep convolutional regression model.

---

### Decision · Program_Chairs · 2023-01-20

**Decision:**

Reject

**Justification For Why Not Higher Score:**

Because of the deficiencies highlighted.

**Justification For Why Not Lower Score:**

Can't go any lower.

**Metareview: Summary, Strengths And Weaknesses:**

The three reviewers had reservations about the novelty of the machine learning model (matrix factorization methodology appears to be highly standard), clarity of the technical contributions, description of the modeling details, and the choice of performance measure. This paper requires a significant rewriting. The scores provided by the reviewers were unanimously below average, hence the rejection decision.

**Summary Of Ac-Reviewer Meeting:**

NIL